# The Unprecedented Role of 3D Printing Technology in Fighting the COVID-19 Pandemic: A Comprehensive Review

**DOI:** 10.3390/ma15196827

**Published:** 2022-10-01

**Authors:** Y. C. Niranjan, S. G. Channabasavanna, Shankar Krishnapillai, R. Velmurugan, A. Rajesh Kannan, Dhanesh G. Mohan, Sasan Sattarpanah Karganroudi

**Affiliations:** 1Department of Mechanical Engineering, Indian Institute of Technology Madras, Chennai 600036, India; 2Department of Mechanical Engineering, Sri Jayachamarajendra College of Engineering, JSS Science and Technology University, Mysuru 570006, India; 3Department of Aerospace Engineering, Indian Institute of Technology Madras, Chennai 600036, India; 4Department of Mechanical Engineering, Hanyang University, 55, Hanyangdaehak-ro, Sangnok-gu, Ansan-si 15588, Korea; 5Institute of Materials Joining, Shandong University, Jinan 250061, China; 6Department of Mechanical Engineering, Université du Québec à Trois-Rivières, Trois-Rivières, QC G9A 5H7, Canada

**Keywords:** COVID-19, 3D printing, additive manufacturing, medical applications, open-source files, innovation

## Abstract

The coronavirus disease 2019 (COVID-19) rapidly spread to over 180 countries and abruptly disrupted production rates and supply chains worldwide. Since then, 3D printing, also recognized as additive manufacturing (AM) and known to be a novel technique that uses layer-by-layer deposition of material to produce intricate 3D geometry, has been engaged in reducing the distress caused by the outbreak. During the early stages of this pandemic, shortages of personal protective equipment (PPE), including facemasks, shields, respirators, and other medical gear, were significantly answered by remotely 3D printing them. Amidst the growing testing requirements, 3D printing emerged as a potential and fast solution as a manufacturing process to meet production needs due to its flexibility, reliability, and rapid response capabilities. In the recent past, some other medical applications that have gained prominence in the scientific community include 3D-printed ventilator splitters, device components, and patient-specific products. Regarding non-medical applications, researchers have successfully developed contact-free devices to address the sanitary crisis in public places. This work aims to systematically review the applications of 3D printing or AM techniques that have been involved in producing various critical products essential to limit this deadly pandemic’s progression.

## 1. Introduction

COVID-19 is an infectious disease caused by a recently discovered coronavirus named Severe Acute Respiratory Syndrome Coronavirus 2 (SARS-CoV-2) [1]. Most COVID-19-infected people experience mild to moderate respiratory illness and can recover without any special medical treatment. Those with pre-medical histories such as diabetes, cancer, and cardiovascular and chronic respiratory diseases are more likely to develop severe illness [1]. Almost every aspect of human life has been affected by the COVID-19 pandemic [2]. As of 31 July 2022, more than 574 million people have been infected by COVID-19, and 6.3 million deaths have been reported globally [3]. Various parts of the world faced the brutal mutations of the coronavirus in the form of waves of cases surging to newer highs than ever at different times [4,5,6,7]. It has been well documented that the very first few cases of COVID-19 were reported in Wuhan, China, as early as December 2019 [4,8,9]. Since then, the novel coronavirus has spread to over 180 countries [8]. The World Health Organization (WHO), on 11 March 2020, declared COVID-19 a pandemic [10].

During the early months of the first outbreak, most countries were forced to be under lockdowns to prevent further outbreaks [11]. This led to an unprecedented situation in which even the best manufacturing industries started failing to produce and dispatch modest protective gear to limit the spread of the virus [12]. Trade and transport restrictions, border controls, production disruptions, and quarantines have greatly affected the global supply chain of critical medical needs [13]. A study stated that about 35% of the supply chains have already been disrupted, and 53% of the manufacturers have anticipated the change in operations due to COVID-19 [14]. According to the director-general of the WHO, Dr. Tedros Adhanom Ghebreyesus, “Without secure supply chains, the risk to healthcare workers around the world is real. Industry and governments must act quickly to boost supply, ease export restrictions and put measures in place to stop speculation and hoarding. We can’t stop COVID-19 without protecting health workers first” [14]. Ever since the advent of this unprecedented situation due to the COVID-19 pandemic, the world has been seeking rapid solutions to meet life-saving medical requirements. Optimal assistance from emerging technologies, including the Internet of Things, additive manufacturing, cloud computing, artificial intelligence, big data, blockchain, and 5G, was immediately required to effectively improve the collective global efforts in virus tracking, prevention and control, epidemic monitoring, treatment, resource allocation, and vaccine development [12].

Unlike subtractive manufacturing, AM is a layer-wise material addition process where a layer of material is deposited over a previously deposited layer [15,16,17]. AM/3D printing technology has emerged as an alternative and rapid solution for manufacturing PPE and medical devices [15,18]. This review article aims to provide insight into the role of AM/3D printing technology as a lifeguarding technology to combat the unprecedented situation of the COVID-19 pandemic. The report also includes information on how this desktop technology with the digital interface responded rapidly to tackle the coronavirus challenges domestically when the big manufacturing firms closed their doors, considering the health risks associated with their mobilized workforce.

## 2. Methodology

Many research articles, short notes, announcements, letters, and other related data from internet sources were initially collected. A detailed literature survey was conducted using Scopus, PubMed, Cochrane, and Google Scholar databases with keywords “COVID-19 and additive manufacturing”, “Coronavirus and additive manufacturing”, “technology and COVID-19”, “COVID-19 and 3D printing”, “Coronavirus and 3D printing”, “3D printing PPE”, and “3D printed isolation wards”. The authors used PubMed and the Cochrane databases to collect the literature on 3D printing in medical applications and Scopus and Google Scholar databases to identify literature related to 3D printing in non-medical applications and the construction of isolation wards. Next, the collected databases were scrutinized to organize the data into three sections, namely 3D printing in medical applications, 3D printing in non-medical applications, and 3D printing in the construction of isolation wards. Finally, the irrelevant articles and data were ignored from the databases, retaining the related ones. All these associated articles and internet source data were combined in a resourceful manner to provide insight into the role played by 3D printing/AM to fight against the novel coronavirus. A detailed workflow chart of the methodology followed is shown in Figure 1.

## 3. COVID-19 and 3D Printing

AM/3D printing is a prominent technology that can provide rapid solutions in its field of application. The significant AM features include rapid response, remote manufacturing, flexibility, decentralized manufacturing, and customization [19,20,21]. The decentralized manufacturing nature of 3D printing can bring out the local microgrids of 3D-printing-based manufacturing firms leading to on-demand manufacturing [22]. These microgrids can supply critical parts locally during severely disrupted supply chains [20]. In addition, AM offers minimal assembly and post-processing steps to deliver the finished products without requiring a mature supply chain and extensive logistics [23]. AM currently features the minimum possible time between thought to product. Flexibility and desktop manufacturing capabilities associated with 3D printing technology have answered the problems arising from global supply chain disruptions [24]. Figure 2 illuminates the application areas of 3D printing during the COVID-19 pandemic.

The era of AM started with limited practical values, including prototyping. Nevertheless, it has grown into a mainstream manufacturing technique with wide acceptance in numerous prominent industries, including automotive, aerospace, defense, and healthcare. In the last decade, the evolution of technology for medical applications has followed the paths of both imagination and problem-solving [25]. Digital technologies such as cloud and digital communication channels have facilitated individual designers to develop open-source 3D models ready-to-be-sent in a 3D printer format. Open-source models have turned out to be a boon during this pandemic [26,27]. The open-source files can be accessed and used by any individual or firm to respond to their requirements. This has harnessed volunteerism in many global 3D printing firms, universities, and other independent groups to provide emergency needs for local hospitals and frontline workers [18]. Aware of AM’s flexibility in producing life-saving supplies, the hobbyists and functional 3D printing firms/labs worldwide have been working collectively on a common goal of safeguarding humankind [27,28]. The Food and Drug Administration (FDA) has emphasized that the 3D printing community response and non-traditional manufacturing have helped to overcome the shortages of medical supplies during this pandemic, in the forms of PPE, medical gear, and other 3D-printed accessories in many parts of the world [29]. Products manufactured by prominent manufacturing industries/organizations with 3D printing facilities before and during the pandemic are given in Table 1 [30,31,32,33,34,35]. According to a report, to address the lack of PPE and other medical requirements, the FDA has attempted to work in partnership with Veterans Affairs (VA), the National Institute of Health (NIH), and America Makes to support non-traditional manufacturing techniques, likely 3D printing. These collaborations have resulted in more than 348,000 3D-printed facemasks and 500,000 3D-printed face shields for healthcare providers and others in need since March 2020 [29].

## 4. AM/3D Printing in Medical Applications during COVID-19

The lack of medical equipment and its components has resulted in a globally soaring COVID-19 mortality rate [13]. In addition, a significant strain has been placed on PPE supplies required to protect the healthcare workers treating critically ill patients [36]. The medical requirement shortages are primarily due to the lack of preparedness in producing or stocking enough supplies for sudden and overwhelming demands. The failures in the medical system to provide tests, care, and protection increased with the growing numbers of infected people [13,18,37]. Suppliers could not meet the rapid demand for medical equipment and PPE caused by the contemporary outbreaks of COVID-19 worldwide. Flanagan et al. [25] from LSU Health Sciences Center Shreveport, United States, requested 3D printers of the world to unite together to keep frontline workers safe in the collective fight against COVID-19. A Few reports from Italy depicted the ugliest side of the pandemic, where life-saving ventilators were allotted based on the best chances of survival and lottery systems [38]. Medical equipment crucially used to diagnose and treat COVID-19-infected patients are diagnostic test kits, PPE, and ventilators [18]. This section overviews how 3D printing has significantly provided rapid solutions to the shortage of global medical requirements.

### 4.1. AM/3D-Printed PPE

PPE acts as a shield between users (generally healthcare professionals) and pathogens. All healthcare professionals must use PPE to reduce the spread of pathogens during pandemic-like situations [39]. PPE is subjected to the FDA enforcement guidelines, but the FDA relaxed its guidelines, citing an attempt to help greater availability of PPE during the public health emergency. The new FDA guidelines for the COVID-19 pandemic do not object to the distribution of improvised PPE as long as they do not cause any “undue risk” [40]. With the FDA’s relaxed guidelines on PPE regulation, there has been a clear indication of the need for 3D-printed PPE [41]. Due to AM’s versatility, Ford Motor Company could 3D print PPE at its Plymouth, Michigan, plant, producing roughly one million face shields per week [42]. Czech-based Tech giant Prusa Research began sharing open-sourced face shield designs, allowing anyone with a 3D printer to download and use the design file to print [41]. Many hobbyists have 3D-printed PPE to help local hospitals with the severe shortages [42]. For the first time, the 3D printer home-user community has joined hands to produce PPE in large numbers. It has been proven that more than 180,000 users worldwide could produce up to 6 face shields in 10 h, each on average, depending on the designs and capabilities of the printers. Greece, having about 500 printers, could produce more than 6000 face shields in a single day, and this was enough to equip its nurses, doctors, rescuers, and staff working in contact with patients [39].

#### 4.1.1. AM/3D-Printed Masks and Face Shields

A facemask is simply a wearable cover acting as a physical barrier to avoid contracting pathogens through the nose and mouth [13,36]. A surgical mask is a loose-fitting disposable face cover. According to the FDA guidelines, masks like N95 and KN95 must have a close fit and seal around the nose and face [43]. The Centers for Disease Control and Prevention (CDC) recommends using N95 masks for healthcare workers in contact with COVID-19 patients and individuals with symptoms [44]. However, these masks are not recommended for multiple reuses, as their filters are inseparable and must be disposed of properly after use. Using low-cost Fused Deposition Modeling (FDM) printers, members of the global 3D printing community have designed and printed reusable facemask frames with insertable filters [36]. Claire et al. [45] printed 50 reusable masks for the Midwestern Trauma Center and made design files available online. These researchers used polylactic acid (PLA) in an Ultimaker S3 FDM printer to produce reusable masks. GrabCAD, a large online community of designers, professionals, manufacturers, and students, have come up with the designs of reusable facemasks, as revealed in Figure 3, and allowed their designs to be open-source [46]. A few available open-source 3D printable files for hobbyists or professionals are provided in Table A1.

FDM printers are widely used to produce facemasks. To make stereolithography desktop printers contribute actively, 3D Systems, a pioneer and the founder of SLA, have designed a facemask comprising a facemask body and filter cover, as shown in Figure 4. The company 3D Systems call this mask a Stopgap facemask, and have recommended using DuraForm ProX PA, PA 1101, and PA 2200 materials in their detailed instructions [48]. In California, a non-profit organization called Maker Nexus has used its 3D printers and laser cutters to produce masks for local hospitals using the open-source Prusa 3D-printed shield design [28].

Medical face shields are head-mounted devices worn as a physical barrier during aerosol-generating procedures such as oropharyngeal suction, respiratory physiotherapy, or intubation [49]. Numerous kinds of face shields fitting different head sizes were 3D printed for nursing homes and regional hospitals during the pandemic [50]. Dina et al. [51] 3D printed face shield frames using biopolymer PLA with a low-cost Prusa FDM printer and assembled them with a transparent sheet. Armijo et al. [52] produced face shields utilizing a combination of 3D printing (FDM) and assembly with commonly available products. In addition, the authors worked on a decontamination protocol to facilitate their reuse. The authors found that the decontamination protocol was highly effective against *E. coli* and *S. aureus* for FDM-printed PLA face shields. An American aerospace manufacturing company Blue Origin and 3D printer manufacturer Carbon have also contributed to producing the face shields [28]. Kantaros et al. [53] utilized pre-existing FDM 3D printing equipment in an academic facility to produce 800 face shields in Greece.

The 3D printing solution provider 3DVerkstan has also responded to the global shortage of face shields by printing face shields, as shown in Figure 5, and made these design files available open source for FDM printers. They claim that the nozzle diameter of up to 1 mm and a layer thickness of 0.5 mm could be used with the polymer filaments such as PLA, CPE, PETG, and ABS for printing face shields [54].

As with facemasks, most face shields were often made using FDM printers. The company 3D Systems has also contributed to designing flexible-type and form-fitting face shields, as shown in Figure 6. They have used their SLA printers and have made their design files available for open access [56]. They recommend medical-grade nylon that is autoclavable and compatible with disinfectant cleaner as a feedstock material for printing reusable face shields [56].

Most hospitals delayed elective arthroplasty surgeries during the later stages of COVID-19 outbreaks, and the helmet systems procured for these surgeries were idle. Duke University Medical Center modified these idle helmets to fit as PPE by using 3D printing techniques. This has set an example for an alternative way to tackle the PPE shortage for frontline workers [57]. Otolaryngologists are generally at higher risk of infection during mouth and nose examinations. Most face shields were not compatible with helmets used during the examination. In this context, Jaime et al. [58] used FDM 3D printers to print adapters to fit the face shield to otolaryngologists’ helmets externally.

#### 4.1.2. AM/3D-Printed Respirator

Powered air-purifying respirators (PAPR) can provide superior protection from the virus [57]. The N95 respirator masks have advantages over surgical and cloth masks. They are tested for the right fit to ensure an adequate seal, to avoid air and tiny droplets entering through the edges of the mask into the breathing zone, and are proven to be more than 95% efficient in filtering airborne particles as small as 0.3 µm. To reduce the impending shortage of N95 masks, the George Washington University Hospital developed reusable respirators with replaceable N95 filters. These respirators could be used with multiple filtration units. Respirators were trial-printed with FDM printable materials such as PLA, ABS, and TPU. The PLA respirators with MERV 16 and MERV 13 sandwich filters were the most valuable options considering their fit, cost, cleaning, and sterilization protocols. Developers also claim that their proposed N95 reusable respirators could be a viable alternative to regular N95 masks [43]. Concerned about the potential shortage of PARPs, which are to be used for intensive and sub-intensive care of patients suffering from COVID-19, Isinnova, an Italian firm, has worked in partnership with Decathlon to convert Decathlons’ Easybreath scuba mask into respirators using connectors [28]. Isinnova called these connectors Charlotte valves, and the 3D printable files can be downloaded from their website directly [47]. The Easybreath scuba mask with a Charlotte valve can be seen in Figure 7. Isinnova also stated that a PLA filament with nozzle temperature of 205–210 °C, bed temperature of 35–50 °C, and a layer thickness of 0.2 mm could be used in FDM printers to print the Charlotte valves [59].

### 4.2. AM/3D-Printed Nasopharyngeal Swabs

The SARS-CoV-2 virus that causes COVID-19 disease can be detected through respiratory samples by reverse transcription-polymerase chain reaction (RT-PCR) or other molecular methods [60]. Nasopharyngeal (NP) swabs are the devices used to capture respiratory mucus and epithelial cells and release the mucus matrix into a transport medium. The transport medium can be analyzed to find viral RNA [61]. The NP swabs must serve three essential functions: (i) must pass through the nasal cavity easily and comfortably; (ii) must collect enough mucus to test for viral RNA; and (iii) should be capable of releasing the collected sample in a manner that will not interfere with the RT-PCR test [62]. The surge in COVID-19 testing has caused an acute global shortage of nasal swabs [63], and access to NP swabs remained a bottleneck for COVID-19 testing in some regions of the world [64]. Using 3D-printed NP swabs to collect nasal samples for COVID-19 testing is feasible, acceptable, and convenient for local production [63]. The FDA classified 3D-printed NP swabs as a Class I, 510(k) Exempt in vitro diagnostic medical device [65]. Formlabs, a 3D printer manufacturer and technology developer, has used its printers to manufacture up to 100,000 nasal swabs daily to tackle these shortages. Printed swabs were shipped to hospitals across the United States that needed early coronavirus detection supplies [28]. Formlabs has used surgical guide resin indigenously developed and specially designed for their printer [66]. Decker et al. [60] printed NP swaps using autoclavable surgical-grade resin (Surgical Guide, Formlabs) and compared them with the conventional Flocked Nasopharyngeal Swabs (FLNP). The 3D-printed swabs displayed statistically identical results to standard FLNP in a head-to-head clinical trial, making them a viable option in COVID-19 testing requirements. These researchers have concluded that 3D printing technology can provide an alternate strategy for swab shortages by facilitating a local solution to FLNP shortages [60]. Sarah et al. [19] printed nearly 2000 swabs using PLA in FDM printers and sterilized them using low-temperature plasma, and these swabs cost them as little as USD 0.05. Nicole et al. [64] designed and printed the NP swab using stereolithography (SLA), as exhibited in Figure 8, and made their design open-access. These designed swabs also absorbed a significant amount of mucus and passed the abrasion and handling tests.

Ian et al. [62] developed lattice bulb NP swabs using Digital Light Synthesis (DLS). These lattice swabs showed efficiency in early clinical trials and met all necessary criteria for an NP swab. Pediatric nasopharyngeal swabs (mini swabs for kids) are a smaller, thinner, and more flexible version of NP swabs used for adults. Pediatric nasopharyngeal swabs faced more severe supply scarcity during the coronavirus pandemic than adult swabs. Starosolski et al. [61], at Texas Children’s Hospital, printed NP swabs using surgical resin. Their work aimed to provide pediatric NP swabs for kids of 1–3 years of age. After sterilization, printed mini swabs were subjected to tensile, torsion, flexural, and fluid absorption tests. Pediatric NP swabs have proven their usability and efficiency with increased mechanical properties while considering a trial on 40 human samples. This work was carried out using the 3D printers adopted by the radiology departments of Texas Children’s Hospital for patient education and surgical planning with the available resources. Olanda et al. [65] evaluated the efficacy of two 3D-printed NP swab designs (Lattice Swab and Origin KXG) for the diagnosis of COVID-19. A total of 70 adult patients (37 COVID-positive and 33 COVID-negative) underwent RT-PCR testing, with a Flocked swab followed by one or two 3D-printed swabs. The authors found high concordance of 3D-printed nasopharyngeal swabs with the control swab results. The authors strongly recommend using 3D-printed NP swabs during the COVID-19 pandemic. This explained the degree of flexibility that 3D printing technology offered in tackling the pandemic. The 3D-printed swabs can also be used to diagnose other common upper respiratory tract pathogens, including respiratory syncytial virus, influenza virus, and Streptococcus pyogenes [63].

### 4.3. AM/3D-Printed Ventilators and Valves

Severely infected COVID-19 patients need ventilators to support respiration [67]. Approximately 2.4% of COVID-19-infected patients need ventilators to support respiration. A mechanical ventilator supports the patient’s respiration by providing positive lung pressure [18]. The first 3D-printed ventilator was developed in Spain [50]. To quicken the design and development process of the ventilators’ critical and intricate components is the need of the hour, and the AM technology has the capability to achieve it [50]. Through integrated manufacturing capabilities of additive manufacturing technology, assembly steps required for ventilators can be minimized. With an agenda to develop critical parts for a low-cost ventilator, the University of Minnesota has collaborated with a 3D printing service provider, Protolabs, to develop essential elements for low-cost ventilators [28].

The game-changing idea of splitting a single ventilator for two or more patients has been proven to be an excellent solution for ventilator shortage [36]. A practical study in 2006 by Greg Neyman [68] indicated that ventilator splitters could be used to support the respiration of four patients with a single ventilator. It has been proposed that a single ventilator can be quickly modified to support four individuals of up to 70 kg for a limited time during an alarming situation such as multiple casualties with respiratory failure [68]. Dependency on the conventional injection molding technique for producing the ventilator splitters takes more than a week. Using 3D-printed ventilator splitters, a single ventilator can accommodate multiple patients in life-saving situations, as demonstrated in Figure 9. Formlabs and Prisma Heath South Carolina have successfully 3D printed these splitters [28]. The ventilator circuit splitter team [69], a medical professional group in the USA, has designed ventilator splitters and made design files open-access. The ventilator circuit splitter team has also discussed print parameters and verification guidelines after printing, according to which the FDM printer with a nozzle diameter of 0.4 mm, 100% infill, 0.2–0.3 mm layer height, and print direction of upside-down Y-direction for circuit splitters and an upright position for limiters can be used with a biocompatible polymer [69].

Ventilator valves are the attachments used to deliver oxygen at fixed concentrations for patients with acute respiratory distress, including COVID-19. In 2020, a hospital in Italy with nearly 250 critically ill COVID-19 patients on ventilator support ran out of ventilator valves needed to connect the patients to the machines. Due to the unexpected increase in demand, the original suppliers could not meet the high demand. Isinnova used 3D printing to respond quickly to the situation and successfully printed ventilator valves [69]. Ventilator valves were prototyped in a short time of 8 h, 100 valves were produced in a day, and each valve costed less than EUR 1 [70,71]. Printed ventilator valves have saved lives, displaying the rapid response capabilities of AM. Isinnova produced ventilator valves prioritizing life-and-death situations over copyrights and medical issues, making them unable to share relevant information and design [58] publicly. The FDA’s recent (26 August 2022) press note addresses using 3D-printed medical device parts and suggests verifying the 3D-printed products’ fit and use before they are used in a clinical setting [72].

AM/3D printing technology was not just limited to addressing the shortage of PPE and other devices during the COVID-19 pandemic. Stephanie et al. [10] have predicted that 3D printing will revolutionize the pharmaceutical industry in drug research, development, and production applicable to the coronavirus. A robust and dynamic drug supply system will be the key to managing future crises. Wen-Kai et al. [23] trust that 3D printing can be used as a decentralized and flexible drug production system with insufficient logistical infrastructure and supply chains. In drug production systems, 3D printing can also be used in disease hot spots with extensive quarantine measures or rural healthcare centers in remote locations [23]. Martin et al. [27] suggested that printing 3D models for planning complex orthopedic, brain, and abdominal tumor surgeries reduces the patients’ admission time in the hospital. This, in turn, reduces the risks of intra-hospital COVID-19 infections and the percentage of bed occupancy. The COVID-19 contagion can be controlled by tracking, monitoring, and early intervention at home. Mohammedhusen et al. [73] developed a bracelet prototype to detect biomedical parameters such as low blood oxygenation or high temperature to provide a home-based solution using 3D printing. These biomedical parameters are instrumental in monitoring a patient with a viral infection. This bracelet is proficient in tracking the number of other bracelets in proximity and can be used to monitor a user’s in-home quarantine. Using Optomec’s Aerosol Jet Technology low-cost printed sensors, researchers at Carnegie Mellon University have developed a device capable of identifying antibodies in 10 to 15 s, and the device is in the trial and testing stage for coronavirus patients [74]. To understand the long-term damage caused by a coronavirus, companies such as Axial3D, Belfast Health, and Social Care Trust have printed lung models. These lung models were prepared using the CT scan data taken on the 14th day of infection and were used to demonstrate the various effects of the virus [74]. A bioprinting company based in San Francisco, Prellis Biologics, explored how synthetic bioprinted lymph nodes can be used to produce fully human COVID-19 antibodies [74]. To reduce the healthcare professionals’ risk while swab testing, robotics researchers from the University of Southern Denmark have developed the world’s first fully automated robot to carry out throat swab tests for COVID-19 [74]

## 5. AM/3D Printing in Non-Medical Applications of COVID-19

Direct contact with commonly touched surfaces, such as elevator buttons, door handles, and computer keyboards, can spread viral diseases [75]. In Paris, to lower the risk of COVID-19 contamination by limiting direct contact, François et al. [75] succeeded in printing hands-free door hooks, openers, and button-pushers, which were later dispatched to Greater Paris University Hospitals and other state institutions. Materialise, a 3D printing and software solution firm, developed several hands-free door openers (Figure 10) and made these designs available open-source [76].

Thermal scanning using handheld infrared thermometers has been the most common method for thermal screening in public places. Personnel involvement in thermal screening has raised concerns, as there is a possibility of violating the safe distance between two or more people [77]. Abuzairi et al. [77] have printed infrared thermometers using 3D printing and made open-source designs to eliminate human dependency on thermal screening. The developers call this 3D-printed infrared thermometer i-Thermowall, which can be seen in Figure 11 [77].

## 6. AM/3D Concrete Printing and COVID-19 Pandemic

The construction sector is one of the most significant contributors to the global economy yet has exhibited notably poor productivity compared to other sectors. It has always been a challenge for the global infrastructure and construction industry to meet the ever-increasing global demands [78]. Post-COVID-19 outbreaks, the demand for isolation wards and quarantine shelters skyrocketed; meanwhile, the disordered and disrupted supply chains posed a more significant challenge for the construction sector. Automation in construction technology has gained prime focus in recent years. The recent trends in 3D printing include its advent in construction technology. AM/3D printing using concert and cob (an earth-based material) has proved its superiority over conventional construction methods. The primary advantages are reduced costs and time required for constructing rapid isolation wards or shelters (350 sq. ft. house can be built within a week’s time) [79,80,81]. In the pre-COVID era, there were few examples of using 3D printing in the construction of houses and shelters [79]. However, in 2018, a group of IIT Madras faculty members collaborated with a startup company Tvasta to build a 350 sq. ft. house in a week inside the university campus [82]. The same Tvasta team in 2020 made a 600 sq. ft. house inside IIT Madras and assured the ability to build homes in 3 weeks. Tvasta says that the green strength of 3D-printed concrete can be achieved in a few minutes, and the overall structure will be ready in 7–10 days [83].

Nonetheless, the literature reveals that the reduced bonding strength between successive layers, increased deformation layers, and dry-shrinkage issues are some of the challenges to be overcome for a better 3D-printed architecture [79,81]. Very few attempts have been made to develop 3D-printed isolation wards/quarantine shelters for coronavirus patients and medical staff. In February 2020, Winsun, a China-based company, 3D printed isolation wards for COVID-19 patients using concrete and recycled materials, each of the structures measuring 10 square meters in area with a height of 2.8 m. It has been claimed that the units were built according to the standards of even withstanding earthquakes and extreme conditions [84].

## 7. Traditional Manufacturing vs. 3D Printing

The comparison between 3D printing and traditional manufacturing in terms of cost, complexity/customization, and production volume is depicted in Figure 12. It is desirable to consider 3D printing in low-volume production and complex or customized parts to be manufactured in significantly less time, as 3D printing eliminates tooling requirements. As explained in the previous sections, 3D printing can also be considered in pandemic-like situations for remote and local manufacturing. Traditional manufacturing is desirable when sufficient time and a mature supply chain exist for large-scale manufacturing.

## 8. Use of AM Technology in India

In India, 3D printing technologies have attracted significant interest from researchers and industries in recent years. Jani et al. [86] reported the application of 3D printing in forensic medicine, odontology, ballistics, and anthropology. Sandhu et al. [87] detailed the materials and 3D printing processes for effective food preparation to stay competitive in the market. Prasad et al. [88] stated the feasibility of manufacturing restorative dentistry, surgical guides, orthodontics, prosthodontics, and implants with the help of computerized tomography scan results. Raichurkar et al. [89] reported the use of a 3D-printed liver model to help a living donor with liver transplantation for the first time in India. Ponnada et al. [90] provided insight into the fabrication of lithium-ion batteries and new energy-storing devices to satisfy the manufacturing requirement of industries. Dehghani et al. [91] examined the cleaner production of textile fabrics with zero-waste environmental design approaches. Dey et al. [92] developed a customized 3D printing machine for preparing cementitious mix and examined the rheological properties using a flow table and vane shear apparatus. Singh et al. [93] reinforced copper with Nylon 6 and Acrylonitrile–Butadiene–Styrene-based thermoplastic polymer nanocomposite filaments to manufacture electronic components via 3D printing technology. Currently, the interest is growing to fabricate complex small- to large-sized components for day-to-day uses and critical applications along with minimal waste and low cost. The 3D printing process has several benefits and drawbacks. The appropriate selection of materials and the 3D printing process are challenging in manufacturing sustainable products.

## 9. Summary and Conclusions

AM/3D printing has allowed manufacturers to contribute differently to the fight against the COVID-19 pandemic. Being a flexible manufacturing technology, AM has proved its efficacy by remarkably answering the global PPE shortage in the shortest possible time. AM’s distributed manufacturing capabilities have truly helped fulfil local requirements when supply chains were severely disrupted. Several open-source 3D printable designs are being created and shared globally for collective efforts. These open-source files are being used by universities, professionals, hobbyists, and several other manufacturing firms to respond and translate themselves to medical and non-medical needs. The translation was and is possible owing to the unique flexible nature of AM. Through 3D-printed splitters, the global shortage of ventilators was reduced to a possible extent. One must be aware that many problems and challenges remain in this field of 3D printing to address medical emergencies. Production by 3D printing in an emergency typically lacks proper quality assurance (QA)/quality control (QC). Even with the same digital model, different printing materials, printers, and printing parameters may lead to the different performance of products. The toxicity of 3D printing materials is another potential issue related to production. Hence, a 3D printing model’s publisher bears responsibility for the model’s validity, and manufacturers or users need sufficient product testing before clinic use. The ISO 13485 standard offers a comprehensive framework for medical devices that ensures product quality and regulatory compliance. In addition, 3D concrete printing has been deployed to remotely construct quarantine shelters during the COVID-19 pandemic, as it requires less human intervention and can construct them in the shortest amount of time. However, 3D concrete printing and related technology have not yet been developed wholly to replace conventional methods, but if the limitations and challenges involved are overcome, 3D concrete printing can be used as a reliable construction technology in the future to handle any subsequent contagion. Apart from these solutions, 3D printing has also been used for operation planning, prototyping, and testing during this pandemic. The scope of AM in this crisis has subsequently increased with the development of much more affordable and reliable 3D printing technology.

The first step in the fight against COVID-19-like outbreaks is to stop their spread by understanding their modes of communication. Infectious diseases transmitted through contact routes need critical attention and quick responses, as they can lead to rapid surges in cases resulting in the catastrophic failure of the healthcare system. Protecting the existing healthcare system is extremely necessary. The extensive research in the context of 3D printing the essentials during the COVID-19 pandemic has increased the preparedness to fight any similar outbreaks effectively. It has become a template for the future. Authors feel that in highly populated countries such as India, prominent education institutes, research laboratories, hospitals, and diagnostic centers can establish Internet of Things (IoT)-enabled 3D printing facilities to interact globally and manufacture locally to tackle challenges such as shortages of PPE and other related issues during times of disrupted supply chains. Further, an open-source computer-based application/portal can be set up by governments/WHO, where proven designs for existing standard models can be made available globally. These global collective efforts are realized to be necessary for controlling and managing COVID-19-like pandemics in the future.

## Figures and Tables

**Figure 1 materials-15-06827-f001:**
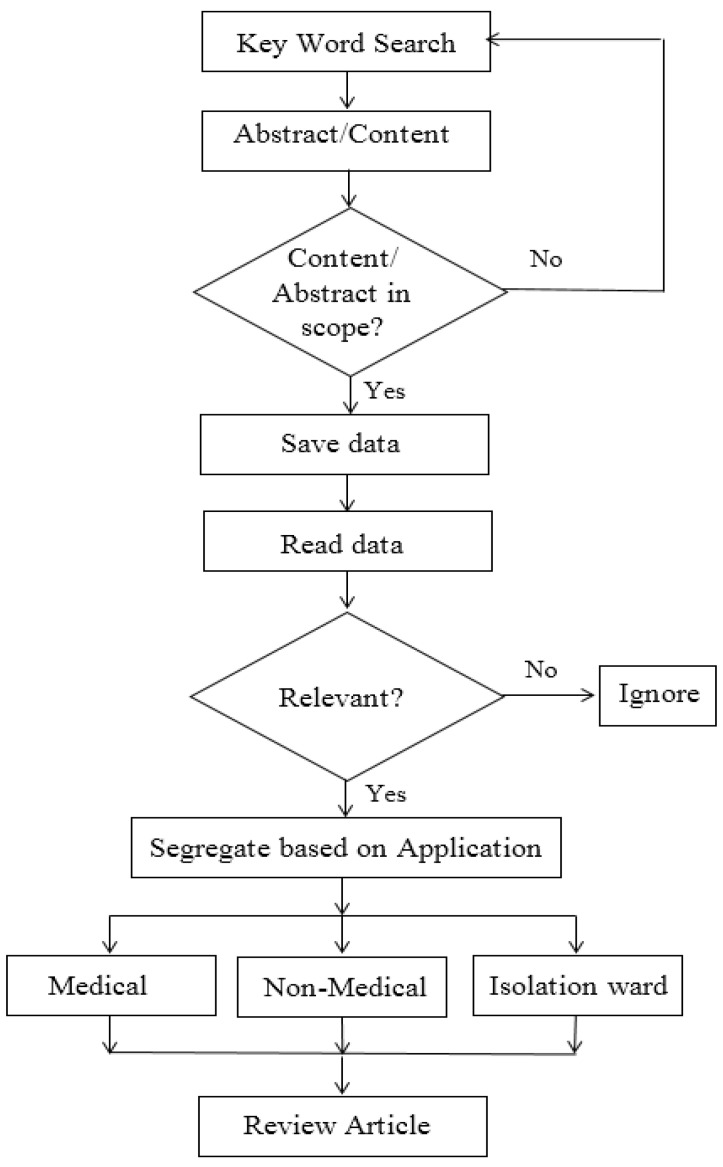
Detailed flow chart of the methodology followed.

**Figure 2 materials-15-06827-f002:**
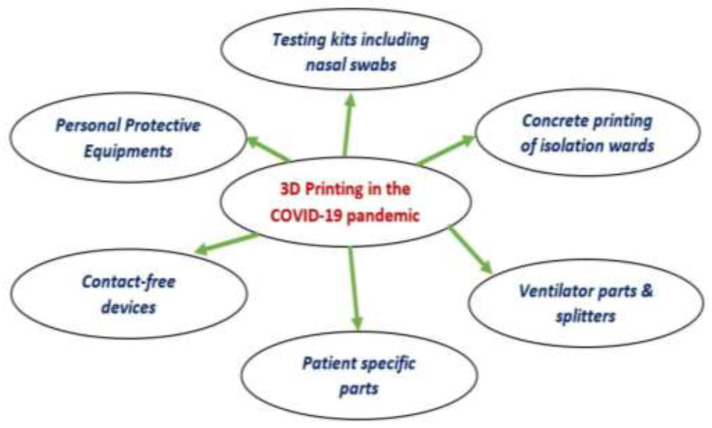
Wide range of applications of 3D printing during the COVID-19 pandemic.

**Figure 3 materials-15-06827-f003:**
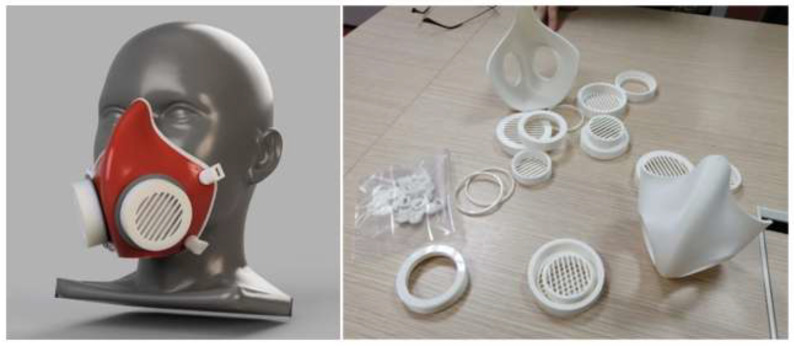
A 3D representation and printed GrabCAD reusable facemask [47].

**Figure 4 materials-15-06827-f004:**
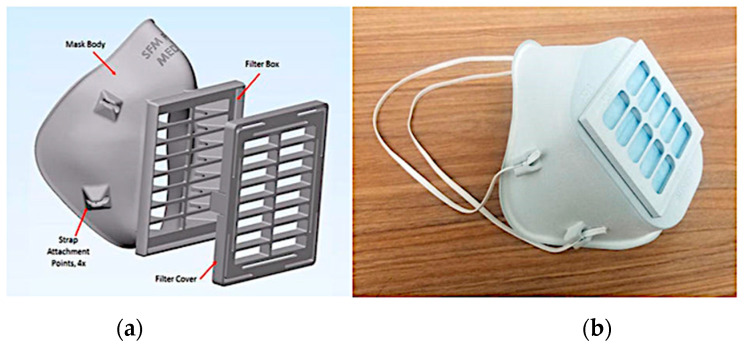
SLA-printed 3D Systems Stopgap facemask: (**a**) CAD representation; (**b**) printed facemask [48].

**Figure 5 materials-15-06827-f005:**
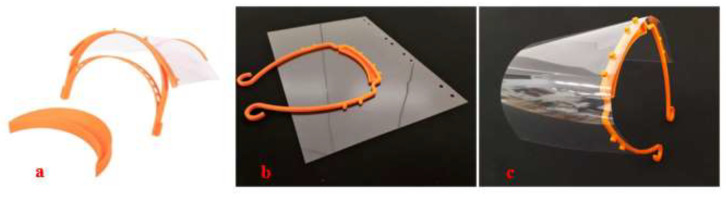
The 3D-printed (**a**) Prusa face shield [55] and (**b**,**c**) 3DVerkstan face shield [54].

**Figure 6 materials-15-06827-f006:**
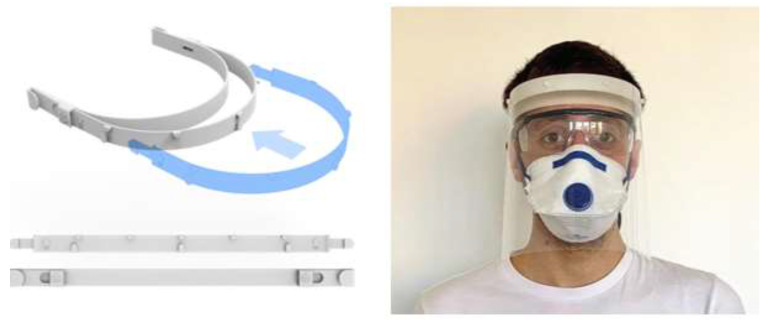
SLA-printed 3D Systems face shield by 3D Systems [56].

**Figure 7 materials-15-06827-f007:**
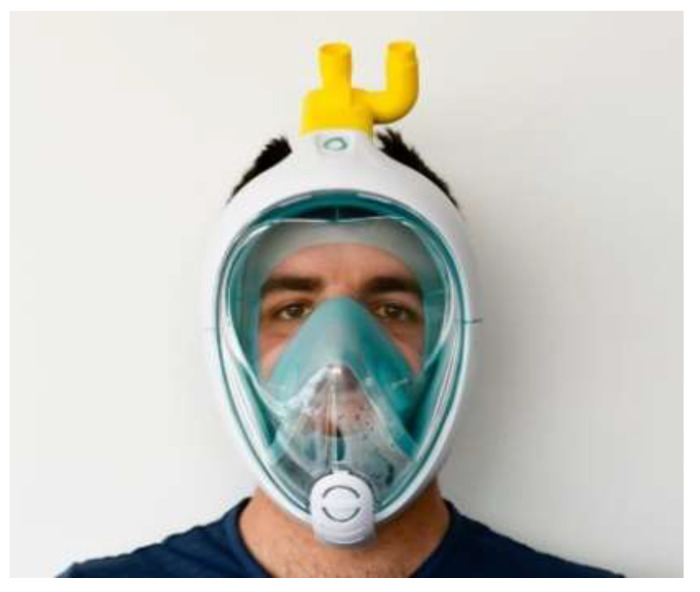
Easybreath scuba mask modified to PARP using Charlotte valve [59].

**Figure 8 materials-15-06827-f008:**
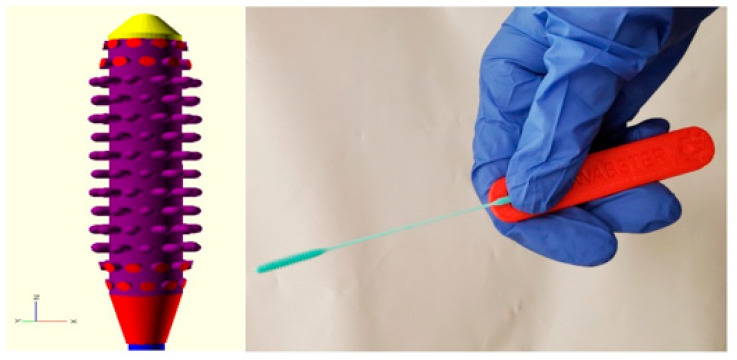
CAD representation and SLS-printed NP swab [64], open access.

**Figure 9 materials-15-06827-f009:**
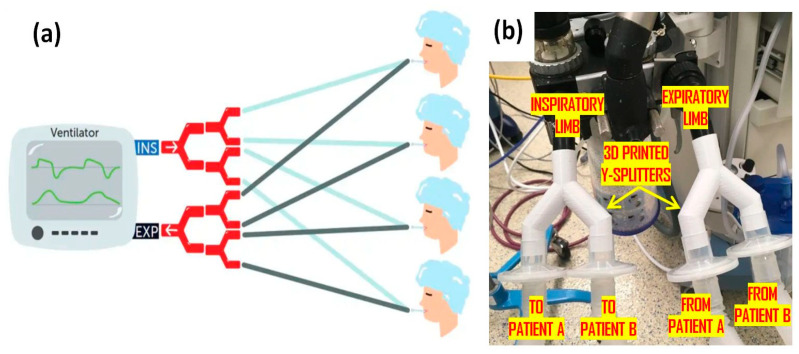
Ventilator splitter: (**a**) schematic representation; (**b**) splitters connected to a shared ventilator [69].

**Figure 10 materials-15-06827-f010:**
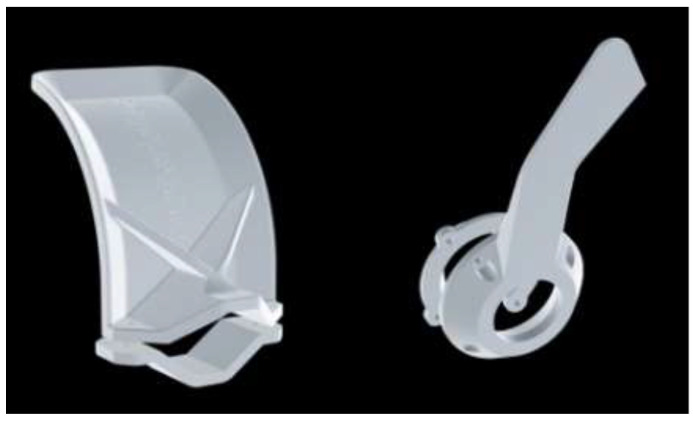
The 3D printable hands-free door opener from Materialise [76].

**Figure 11 materials-15-06827-f011:**
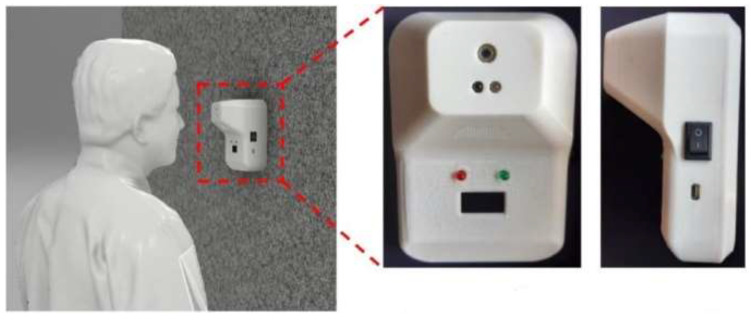
The 3D-printed infrared thermometer (i-Thermowall) [77]; open access.

**Figure 12 materials-15-06827-f012:**
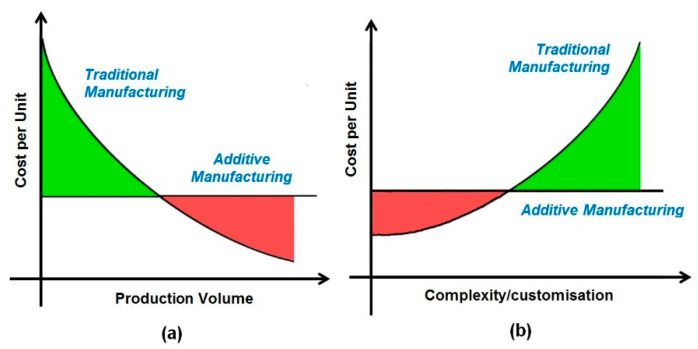
Cost and complexity comparison between (**a**) 3D printing (additive manufacturing) and (**b**) traditional manufacturing [85].

**Table 1 materials-15-06827-t001:** Products produced by prominent manufacturing industries/organizations with 3D printing facilities before and during the pandemic.

Sl. No	Manufacturer/Organization	Domain	Products Manufactured		Ref.
			Before Pandemic	During Pandemic	
1	Airbus	Aerospace	Aircraft components	Medical visors	[30]
2	Ford	Automotive	Automotive components	Respirators, ventilators, facemasks, and face shields.	[31]
3	General Motor	Automotive	Automotive components	Ventilators and their components	[32]
4	Nagami	Product Design	Furniture	Masks	[33]
5	Toyota	Automotive	Automotive components	Face shields and ventilators	[34]
6	Yingchuang Building Technique (Shanghai) Co., Ltd., Shanghai, China	Construction	3D printing architecture	Isolation wards and quarantine shelters	[35]

## Data Availability

Not applicable.

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
