# Peer review of "The Unprecedented Role of 3D Printing Technology in Fighting the COVID-19 Pandemic: A Comprehensive Review"

_materials, 2022, doi:10.3390/ma15196827_

Round 1

Reviewer 1 Report

This paper is a review, so there is a sense of déjà vu to all of the information. I think the content is fine. It would be better if it included the author's opinions, suggestions, and the situation of the use of AM technology in India, if possible.

Author Response

Response to Reviewers' Comments:

Before answering the comments, we like to express our gratitude to the respected reviewer for their time and valuable input. His/Her input helped us improve the manuscript and bring it out as the best. Here are the responses to the respected reviewers' comments and the revisions made to the manuscript.

Reviewer 2 Report

The manuscript attempts to present a review about the role of 3D Printing technology in fighting the Covid-19 pandemic. More specific, the paper presents the cases where 3D Printing was used in order to produce PPEs (Personal Protective Equipment). The paper has very limited scientific value, since there have been tons of very similar papers in the literature. The paper would have a scientific value if it was presented 2 at least 2 years ago. Even as a review, it has limited references that should approach at least 80-100. As a last chance, I urge the authors to include more references.

My points are analytically listed below

Points for consideration:

Point 1: Authors should include more references for the paper to have some marginal value as a review. Include at least, the following.

·         10.1016/j.ajic.2020.07.037

·         10.1016/j.xinn.2020.100056

·         10.1053/j.jvca.2020.04.004

·         10.1007/s43615-021-00047-8

·         10.1016/j.diagmicrobio.2020.115257

·         10.1016/j.acra.2020.04.020

·         10.1016/j.jormas.2020.06.010

·         10.1016/j.ssci.2020.104870

·         10.1016/j.ajem.2020.08.010

· Point 2: In line 73, please insert the following relevant reference as well regarding low cost FFF 3D Printers that were used by the community for producing PPEs.

·  10.5923/j.mechanics.20211001.02

· Point 3: Check all the text for English language mistakes or phrasal mistakes as there are many of them with many sentences not making easily sense, or have an English native speaker check the whole text.

Author Response

(The authors gave the same response as above.)

Reviewer 3 Report

Please review the attached report for comments on the manuscript.

Author Response

(The authors gave the same response as above.)

Reviewer 4 Report

The paper The Unprecedented Role of 3D Printing Technology in Fighting the Covid-19 Pandemic: A Comprehensive Review is well written and comprehensive. The content is relevant and interesting.

Methodology: It may be beneficial to further explain the written text of the methodology for inclusion of papers. Additionally, as the types of manuscripts being reviewed include both engineering and medical it may be beneficial to share the specific criteria and how it varies across disciplines. Health related fields often include Cochrane or PEDro assessment of literature for indicating the quality of the manuscripts, while PRISMA might be appropriate for engineering or manufacturing fields. A short description in the text might be supportive for deeming the relevance 

Printed devices: Other 3D printing companies may have also participated in 3D printing networks for face shields, additional to Prusa. 

Ventilators: The topic for 3D printed ventilators is complex in particular to the use case, risk, efficacy, and reception. Tones on ventilators for COVID-19 patients changed over 2020 additionally. It may be appropriate to add a discussion on these tones and how rapid prototyping helped support supply chain shortage short term gaps. 

Summary and conclusions: this section may be extended to provide recommendations for crisis related manufacturing. Recommendations for distributed manufacturing, regulation, testing, and beyond may be supportive for the article. 

Imagery: the images are generally clear and well made. Figure 9 may be difficult to read. 

Author Response

(The authors gave the same response as above.)

Round 2

Reviewer 2 Report

I would like to thank the authors for their revisions. The manuscript looks far better now, with more references that increase its value as a review. I consent to its publication.